# Clinical and Sociodemographic Profile of Psychomotor Agitation in Mental Health Hospitalisation: A Multicentre Study

**DOI:** 10.3390/ijerph192315972

**Published:** 2022-11-30

**Authors:** María Elena Garrote-Cámara, Vicente Gea-Caballero, Teresa Sufrate-Sorzano, Esther Rubinat-Arnaldo, José Ángel Santos-Sánchez, Ana Cobos-Rincón, Iván Santolalla-Arnedo, Raúl Juárez-Vela

**Affiliations:** 1Care and Health Research Group, Department in Nursing, University of La Rioja, C/Duquesa de la Victoria 88, 26004 Logroño, Spain; 2Research Group on Community Health and Care, Faculty of Health Science, Valencia International University, 46002 Valencia, Spain; 3Society, Health, Education and Culture Study Group, Department of Nursing and Physiotherapy, Faculty of Nursing and Physiotherapy, University of Lleida, 25003 Lleida, Spain; 4Faculty of Medicine, University of Salamanca, 37008 Salamanca, Spain

**Keywords:** mental health, nursing, psychomotor agitation, severe mental disorder

## Abstract

Psychomotor agitation is characterised by an increase in psychomotor activity, restlessness and irritability. People with psychomotor agitation respond by over-reacting to both intrinsic and extrinsic stimuli, experiencing stress and/or altered cognition. The objective of this study is to assess the clinical and sociodemographic profile of psychomotor agitation in patients with severe mental disorders. The study was carried out in Spain by means of multicentre cross-sectional convenience sampling involving 140 patients who had been admitted to psychiatric hospital units and had experienced an episode of psychomotor agitation between 2018 and 2021.Corrigan’s Agitated Behaviour Scale was used to assess psychomotor agitation. The results show that the predominant characteristic in psychomotor agitation is aggressiveness, which is also the most reported factor in patients with severe mental disorder. Patients who also have anxiety develop psychomotor agitation symptoms of moderate/severe intensity. The clinical and sociodemographic profile found in our study is consistent with other studies on the prevalence of psychomotor agitation.

## 1. Introduction

Psychomotor agitation is characterised by an increase in psychomotor activity, motor restlessness and irritability. People with psychomotor agitation respond by over-reacting to both intrinsic and extrinsic stimuli, experiencing stress and/or altered cognition. Among the possible forms of presentation of psychomotor agitation are mental illness, disease of the central nervous system, associated organic pathology and/or substance abuse, among other aetiologies [1,2]. Psychomotor agitation is a behavioural category of motor hyperactivity that leads those affected from it to engage in unproductive, incomplete and repetitive conduct; it does not constitute a specific condition but rather a syndrome that can be present in various pathologies [3,4]. In 2018, professional experts defined psychomotor agitation as “a state where patients cannot remain still or calm, characterized by internal features such as hyperresponsiveness, racing thoughts and emotional tension; and external ones, mainly motor and verbal hyperactivity, and communication impairment” [5]. With the above, psychomotor agitation can be circumscribed as a syndrome with origin in different factors that implies the alteration and lack of coordination in the motor sphere of the patient and multiple signs and symptoms, among which aggressiveness, lack of inhibition, lability and anxiety can be highlighted [6]. During an episode of psychomotor agitation, the patient’s mood can present as nervous, euphoric or angry, with laughing, crying and uncontrollable shouting being common manifestations, which may lead to verbal and/or physical aggressiveness and therefore represent a serious risk for both the patient themselves (auto-aggressiveness) and for their relatives, healthcare personnel and their surroundings in general (hetero-aggressiveness) [7].

Psychomotor agitation is associated with certain risk factors [1]. Demographic factors include being male, being aged younger than 40 years, being single, having a family history of alcoholism or aggressive behaviour, having a low level of education and being of a low socioeconomic level; psychological factors include having a history of conflict with healthcare personnel or other patients, recent stressful life events or involuntary or prolonged admission to hospital; and clinical factors include having a family history of previous episodes of agitation, anxiety, fear, substance abuse, low cooperation in treatment, low-level awareness of illness, cognitive and behavioural disorganisation and positive symptoms, mental retardation, dementia, epilepsy, schizophrenia and comorbidity with personality disorders [8]. A strength of this study is to relate psychomotor agitation to the most prevalent nursing diagnoses in severe mental disorder.

The prevalence of psychomotor agitation in emergency departments is between 10% and 50% in patients with bipolar disorder, schizophrenia and dementia [9,10]. Numerous studies link psychomotor agitation with severe mental disorder, other psychiatric disorders and disorders of medical origin, such as dementia [11,12,13,14]. Agitation of psychiatric origin occurs more frequently in patients with psychotic episodes, schizophrenia, schizoaffective disorder, bipolar disorder in its manic phase and in some personality disorders [15,16,17,18,19]. In Spain, a study showed that 25% of patients with schizophrenia and 15% with bipolar disorder had an annual episode of psychomotor agitation, with a median of two episodes [20,21]. The STAGE study found that almost 5% of its sample in psychiatric emergencies were psychomotor agitation events, 63% being male, mostly with schizophrenia and bipolar. Most episodes of agitation were related to substance abuse, although they were usually associated with mental illness [22].

Psychomotor agitation is a health emergency that is treated as a severe mental disorder, requires immediate professional care and seriously compromises the patient’s safety [11,14,19,23,24,25]. The treatment and handling of patients affected by psychomotor agitation presents major difficulties; motor alteration, lability, disinhibition, a lack of cooperation and the possibility of aggressive behaviour makes it difficult to assist, treat and care for the patient. Lindenmayer [26] proposed four characteristics that are often present in agitated patients: restlessness with excessive or semi-intentional motor activity, irritability, increased sensitivity to external and internal stimuli and an unstable clinical course. Agitation can also be one of the main indicators of imminent and impulsive suicidal behaviour [27,28,29,30]. These features mean that the patient requires immediate attention, as well as quality professional care, but a lack of collaboration from the patient during episodes often delays the obtaining of an adequate psychiatric history and the start of treatment. The assessment of psychomotor agitation is a challenge for health professionals, since adequate assessment is key to effectively managing the patient [31,32]. Psychomotor agitation requires early recognition and appropriate assessment and management to minimise complications and risks for the patient, professionals and their surroundings [2]. One of the most used scales for assessing agitated behaviour is Corrigan’s Agitated Behaviour Scale (ABS), which provides quantitative and qualitative information, allowing to know the severity of agitation and its properties in terms of lability, disinhibition and aggressiveness [33,34]. It has been mostly used in the assessment of psychomotor agitation in mental health settings [13,34,35,36,37]. This evaluation can be completed with the assessment of associated nursing diagnoses, according to the North American Nursing Diagnosis Association (NANDA). Nursing diagnoses are defined as a clinical judgement about the responses of the individual, family or community to actual or potential health problems or life processes. Determining the nursing diagnoses associated with episodes of psychomotor agitation is essential to determining efficient nursing interventions to ensure quality care and provide clinical safety and in the setting and resolution of the case [38]. Nursing diagnoses are fundamental to providing high-quality nursing care and improving clinical organisation and risk management. This leads to better care planning and a better understanding of the aspects that are more difficult to assess, thus reinforcing the commitment between nursing theory, practice and education, based on the best scientific evidence [39]. Mental health nurses are trained to provide specific care to the patient, family and community facing mental health problems via the promotion of health and prevention of illness, providing the necessary care through an interpersonal relationship as a therapeutic instrument [40]. Given how important it is to better understand the characteristics of episodes of psychomotor agitation, to aid its assessment, the objective of this study was to assess the clinical and sociodemographic profile of psychomotor agitation in patients with severe mental disorders.

## 2. Materials and Methods

### 2.1. Data, Instrument and Sample

The study was carried out in La Rioja, Spain, by means of multicentre cross-sectional convenience sampling. The sample comprised 140 patients who had been admitted to psychiatric hospital units in the Rioja Health Service who met the following inclusion criteria: patients diagnosed with a severe mental disorder; older than 16 years; admitted to units within the Rioja Health Service’s mental health network; having experienced an episode of psychomotor agitation between 2018 and 2021; in the discharge process (the medical discharge report has been prepared) and not legally disabled; having no perceptual and/or cognitive alteration preventing them from understanding the nature of their disorder, the clinical reasons for their admission or the objectives of this research and the data used in it.

Corrigan’s Agitated Behaviour Scale (ABS) was used to assess psychomotor agitation. The ABS is a hetero-applied tool that assesses 14 items on a Likert-type scale [13,33,34,37]. The scale was completed by nursing staff using a Likert-type scale with 4 levels of intensity, from 1 (absence) to 4 (extreme degree). This scale is composed of 14 items grouped into three dimensions: disinhibition (1, 2, 6, 7, 8, 9 and 10), aggressiveness (3, 4, 5 and 14) and lability (11, 12 and 13) [33]. The sum of the 14 items give us the result of the severity of psychomotor agitation, with a minimum of 14 points and a maximum of 56 points, the latter figure corresponding to greater severity. The individual assessment of the dimension establishes whether the agitation is manifested more by the disinhibition dimension (intermediated score which ranges between 7 and 28 points) or aggressiveness dimension (intermediated score which ranges between 4 and 16 points) and lability dimension (intermediated score which ranges between 3 and 12 points). In the overall assessment of agitation, intermediated scores which range between 14 and 27 points are considered minor agitation, intermediated scores which range between 28 and 41 are considered moderate agitation and intermediated scores which range between 42 and 56 are considered severe psychomotor agitation. With regard to the disinhibition dimension, intermediated scores which range between 7 and 13 indicate a low predominance, intermediated scores which range between 14 and 20 indicate medium predominance and intermediated scores which range between 21 and 28 indicate a high predominance of disinhibition dimension in the psychomotor agitation episode. With regard to the aggressiveness dimension, intermediated scores which range between 4 and 7 indicate a low predominance, intermediated scores which range between 8 and 11 indicate medium predominance and intermediated scores which range between 12 and 16 indicate a high predominance of aggressiveness in the psychomotor agitation episode. With regard to the lability dimension, intermediated scores which range between 3 and 5 indicate a low predominance, intermediated scores which range between 6 and 8 indicate medium predominance and intermediated scores which range between 9 and 12 indicate a high predominance of lability in the psychomotor agitation episode [33,34]. Various studies have shown the sound psychometric properties of the English version, with a Cronbach’s alpha of between 0.801 and 0.921 [13,33,34,41]. The recent study of the Spanish version of the scale shows complete consistency (0.9 Cronbach’s alpha), as well as the consistency of each of the three domains of which it is made up [35] (0.8 disinhibition, 0.8 aggressiveness and 0.7 lability), with similar values to those found in the original version [33].

This scale, in its Spanish version, showed that the dimensions disinhibition, aggressiveness and lability, in the result of Cronbach’s alpha, gave lower scores than that of the total scale, concluding that the total weighting of the scale is the best representation of agitation. In this version of the scale, the three dimensions account for more than 64% of the total variance, exceeding the 50% of the original version.

In addition to the data provided by the ABS scale, the predictive variables in this study included sociodemographic data (age and sex), the main medical diagnosis behind the patient’s admission and associated North American Nursing Diagnosis Association (NANDA) nursing diagnoses such as anxiety (00146), risk of suicide (00150), acute confusional disorder (00128) and auditory perceptual disorder (00122) [42], as these are the nursing diagnoses most closely associated with episodes of psychomotor agitation.

### 2.2. Statistical Analyses

The sociodemographic and clinical variables were analysed using descriptive statistics such as mean and standard deviation for quantitative variables and frequencies for categorical variables. Age, sex and clinical diagnosis of mental illness were used as sociodemographic variables. In addition, median, asymmetry and kurtosis were used to describe the responses to the overall score items on the ABS scale. In the bivariate analysis, Pearson’s chi-squared tests were used to evaluate significant differences (*p* < 0.05) in psychomotor agitation in patients with severe mental disorders for qualitative predictor variables; depending on the type of distribution, we used Student’s *t*-test and ANOVA or Mann–Whitney U-test and Kruskal–Wallis test and related post hoc tests when necessary. Following the central limit theorem, we established that samples of variable categories with a size > 30 patients would approximately follow a normal distribution [43]. For ordinal logistic regression, we used the multivariate linear analysis model to determine those predictor variables that influence the outcome of psychomotor agitation. As for the predictive variables related to the different levels of intensity of psychomotor agitation, minor, moderate and severe, the following variables were used: anxiety, suicide risk, acute confusional disorder, auditory perceptual disorder and gender. Variables with *p* < 0.05 in the bivariate analysis were discarded and variables with statistical significance were included in the final model. The statistical software used was IBM SPSS Statistics 25.0 [44].

### 2.3. Ethical Considerations

The collection of data was anonymous and no personal data nor data which could identify the participant were collected. The information was processed on a confidential and anonymous basis, following the European Parliament’s General Data Protection Regulation (EU) 2016/679 and Spanish Organic Law 3/2018. The study was approved by the La Rioja Biomedical Research Centre’s Ethics Committee: CEImLar (La Rioja Drugs Research Ethics Committee) reference P.I. 467. The researchers declare that they have no ethical, moral or legal conflicts of interest, and have not received financial compensation of any kind. All participants voluntarily gave their informed consent.

## 3. Results

Table 1 illustrates the sociodemographic properties. In the overall sample, 52.9% were men and 47.1% were women, with an average age of 45.6 years. A total of 60.7% had schizophrenia and other psychotic disorders as their underlying pathology, with a gender distribution of 60% men and 40% women. Depression was present in 9.3% of the sample (30.8% men and 69.2% women) and mania in 2.1% (100% women). Finally, 8.6% had bipolar disorder (33.3% men and 67.7% women) and 10.7% had a personality disorder, with a distribution of 40% men and 60% women.

Appendix A shows the mean, standard deviation (SD), skewness and kurtosis for each of the items of the Spanish version of the Corrigan ABS psychomotor agitation scale. For the most part, the distribution of the items is normal, without excessive skewness and kurtosis. The highest weights were found in item 2 “Impulsive, impatient, tolerates pain or frustration poorly” and item 3 “Uncooperative, does not let them take care of him, demanding”. The lowest scores were recorded in item 7 “Pulls the tubes or ties on the bed” and item 13 “Cries or laughs easily and excessively”.

The mean, standard deviation and median, minimum and maximum values for the total value of the scale and for each of the dimensions of the psychomotor agitation scale are shown in Table 2. The sum total of the items in the scale showed a median (p50) of 37, a score which represents moderate/high agitation on average. With regard to the analysis dimensions, the predominant characteristic in psychomotor agitation is aggressiveness, with a median (p50) score of 12. Lability also has a medium/high predominance, while disinhibition has average predominance.

As presented in Table 3, a majority of patients (64.29%) experienced moderate psychomotor agitation, and 26.42% experienced severe psychomotor agitation. The aggressiveness dimension is more associated with psychomotor agitation in mental health patients; 40.71% of patients experienced high levels of aggressiveness during the episode of agitation. The other dimensions reached moderate levels of intensity; the patients showed moderate levels of disinhibition in 60.71% and of lability in 61.43% of cases.

The statistically significant association between the predictive variables and the level of psychomotor agitation and its dimension is shown in Table 4. Patients who also have anxiety develop psychomotor agitation symptoms of moderate/severe intensity; none of the patients who presented anxiety experienced minor symptoms of agitation, and these data are of statistical significance (*p* = 0.034). There is a statistically significant relationship between anxiety and the presence of the disinhibition dimension in episodes of psychomotor agitation (*p* = 0.024); among patients who also have anxiety, the predominance of moderate/severe disinhibition is more frequent. The presence of anxiety is also associated with the onset of aggressiveness dimension at moderate levels (*p* = 0.029) and with moderate/severe levels of the lability dimension (*p* = 0.040). In relation to the suicide risk variable, the symptoms of psychomotor agitation associated with this variable are predominantly of minor intensity (*p* = 0.027); this variable is not very present in moderate- or severe-level psychomotor agitation. Patients who also have acute confusional disorder developed symptoms of psychomotor agitation with the presence of severe levels of the disinhibition dimension. The relationship is shown to be statistically significant (*p* = 0.046), and the levels of disinhibition increase with the acute confusional disorder variable. A statistically significant relationship (*p* = 0.009) is observed between the levels of the aggressiveness dimension present in an episode of psychomotor agitation and the presence of auditory perceptual disorder in the patient; the presence of this variable is associated with moderate levels of the aggressiveness dimension. In relation to the sex variable, statistically significant differences (*p* = 0.049) were found, exclusively with the presence of the lability dimension. The levels of lability in episodes of psychomotor agitation are of greater intensity in women than in men.

An assessment using multivariate analysis of the effect of the variables on the risk of psychomotor agitation is shown in Table 5. This analysis shows that a nursing diagnosis of auditory perceptual disorder is correlated to a statistically significant degree with high levels of psychomotor agitation (OR: 4.75; *p* < 0.05) and with the onset of psychomotor agitation symptoms with a predominance of the aggressiveness dimension (OR: 3.03; *p* < 0.05). The nursing diagnosis of anxiety shows a statistically significant correlation with high levels of aggressiveness in the development of psychomotor agitation symptoms (OR: 1.99; *p* < 0.05). The multivariate analysis of the nursing diagnosis of confusional disorder shows a statistically significant correlation with levels of psychomotor agitation considered to be minor, where the disinhibition dimension is predominant (OR: 0.27; *p* < 0.05).

## 4. Discussion

This study explored the clinical and sociodemographic profile of psychomotor agitation in mental health patients. The clinical and sociodemographic profile found in our study is consistent with the study by Hart et al. on the prevalence, characteristics and implications of agitation in patients admitted to hospitals with mental illness, where the mean age was 44.81 years and 49.8% of the episodes of agitation were in men, without age or sex being associated with agitation [15]. This study by Hart et al. linked the highest levels of agitation with psychotic disorders and bipolar mania, and they were associated with a longer time spent admitted to the hospital.

Our study records severe psychomotor agitation in more than 25% of admitted patients and moderate psychomotor agitation in more than 60% of admitted patients who had an episode of psychomotor agitation. These data are in line with other studies such as Sacchetti et al., where the prevalence of moderate psychomotor agitation was 40.5% and the prevalence of severe psychomotor agitation was 23.7%, observing that the latter were younger, with a more recent onset of illness and greater recent consumption of substances; this study pertained specifically to patients with schizophrenia [45].

The aggressiveness dimension is more associated with psychomotor agitation in mental health patients; more than 90% of patients experienced moderate/high levels of aggressiveness during the episode of agitation. This association is present in the reviewed scientific literature. In the Sakanaka et al. study [46], symptoms of excitement from the PANSS five-factor model and frontal dysfunction were significantly associated with high aggressiveness, and it also emerged that impulsive traits of aggressiveness are less relevant than other factors [46,47,48,49,50].

Our results show a significant association between moderate/severe symptoms of psychomotor agitation and anxiety. In this regard, other reviewed studies reveal that moderately or severely agitated patients have higher scores in the anxiety sub-scale, in the total Positive and Negative Syndrome Scale (PANSS) and in Emsley anxiety factors [45,46,47,48]. According to the conclusions of the study carried out by Alderson-Day et al., anxiety is present in 62.5% of patients with a severe mental disorder who also have auditory perceptual alteration, where participants reported voices associated with negative emotions [49]. In this regard, our study associates the nursing diagnosis of auditory perceptual disorder and the associated diagnosis of anxiety with high levels of psychomotor agitation and the aggressiveness dimension. Anxiety-associated behaviours originate when the situation exceeds the individual’s ability to respond, and may include tachypnea, crying or repeated hand rubbing [50]. Anxiety is particularly prevalent in auditory perceptual disorder, in both clinical and non-clinical populations, and is the most prominent emotion during hallucinations [51,52,53]. The anxiety intensity reports from another study associated with psychomotor agitation reveal that they exceeded benchmark levels [54].

The report on agitation and anxiety studied specifically in patients with Alzheimer’s disease proposed the hypothesis that agitation may be an expression of anxiety. The results confirm the theory that both entities, agitation and anxiety, are distinct entities, and that furthermore, anxiety was not related to the future development of an agitation episode in Alzheimer patients at the beginning of the research, although a relationship between the two was found later. [55].

## 5. Conclusions

This study allows us to quantify the level of psychomotor agitation as moderate–severe in mental health patients. At the qualitative level, out of the disinhibition, aggressiveness and lability dimensions that the ABS scale assesses, it is the aggressiveness dimension that is more associated with psychomotor agitation in these patients; the patients experienced high levels of aggressiveness during the episode of agitation. The associated nursing diagnoses of anxiety and auditory perceptual disorder are found to be predictive factors for the severity of the episode of agitation. The determination and assessment of anxiety and perception disorders in patients with severe mental disorder during hospitalisation, as well as intervention at this level, can prevent the onset of episodes of psychomotor agitation and reduce their intensity, preventing serious complications associated with these symptoms.

## 6. Limitations and Strengths

One of the main strengths of this study is its implication for health professionals in the standardisation of an assessment method for episodes of psychomotor agitation in mental health hospital units, as well as its contribution of results in an area that is currently little researched, namely psychomotor agitation of psychiatric origin. Another significant contribution of this research is a better understanding of clinical and sociodemographic characteristics of patients with psychomotor agitation and presentation and use of assessment tools which enhance nursing care.

As an implication for clinical practice, knowing the clinical profile of these episodes improves the healthcare provided to the patient and family. The study’s limitations include the sample collection obtained through convenience sampling, related to the complexity of working in an environment that is difficult to access, where achieving voluntary and active patient participation becomes more problematic, and more so when it comes to analysing behaviours associated with a mental disorder which is unfortunately still subject to high levels of stigmatisation.

## Figures and Tables

**Table 1 ijerph-19-15972-t001:** Sociodemographic properties (*n* = 140).

Variables		%	*n*
Age	<18	1	0.7
	18–30	21	15
	31–50	69	49.3
	51–65	28	20
	66–79	17	12.1
	>80	4	2.9
Sex	Women	66	47.1
	Men	74	52.9
Mental Illness	Schizophrenia and other psychotic disorders	85	60.7
	Depression	13	9.3
	Mania	3	2.1
	Bipolar	12	8.6
	Personality disorder	15	10.7
	Others	22	8.6

**Table 2 ijerph-19-15972-t002:** Mean, standard deviation and median of the total level of psychomotor agitation and its dimensions.

Dimension	Mean	SD	Median P50	Min	Max
Disinhibition	17.9	±0.5	18	7	28
Aggressiveness	11.6	±0.3	12	4	16
Lability	7.7	±0.3	8	3	12
TOTAL	37.4	±0.8	37	14	56

**Table 3 ijerph-19-15972-t003:** Psychomotor agitation intensity.

Dimension	Minor	Moderate	Severe
Disinhibition	17.86%	60.71%	21.43%
Aggressiveness	8.58%	50.71%	40.71%
Lability	16.43%	61.43%	22.14%
TOTAL	9.29%	64.29%	26.42%

**Table 4 ijerph-19-15972-t004:** Predictive variables (frequencies and percentages).

**Anxiety (*p* = 0.034)**	**Psychomotor Agitation Intensity**		
	**Minor**	**Moderate**	**Severe**
Yes	0%	35.56%	29.73%
No	100%	64.44%	70.27%
**Risk of Suicide (*p* = 0.027)**	**Psychomotor Agitation Intensity**		
	**Minor**	**Moderate**	**Severe**
Yes	15.38%	1.11%	5.41%
No	86.62%	98.89%	94.59%
**Anxiety (*p* = 0.024)**	**Disinhibition Dimension**		
	**Minor**	**Moderate**	**Severe**
Yes	8.00%	36.47%	33.33%
No	92.00%	63.53%	66.67%
**Acute Confusional Disorder (*p* = 0.046)**	**Disinhibition Dimension**		
	**Minor**	**Moderate**	**Severe**
Yes	0%	9.41%	20%
No	100.00%	90.59%	80%
**Anxiety (*p* = 0.029)**	**Aggressiveness Dimension**		
	**Minor**	**Moderate**	**Severe**
Yes	25%	40.85%	19.3%
No	75%	59.15%	80.7%
**Auditory Perceptual Disorder (*p* = 0.009)**	**Aggressiveness Dimension**		
	**Minor**	**Moderate**	**Severe**
Yes	0%	12.68%	0%
No	100%	87.32%	100%
**Anxiety (*p* = 0.040)**	**Lability Dimension**		
	**Minor**	**Moderate**	**Severe**
Yes	13.04%	30.23%	45.16%
No	86.28%	69.77%	54.84%
**Sex (*p* = 0.049)**	**Lability Dimension**		
	**Minor**	**Moderate**	**Severe**
Yes	48.73%	60.47%	35.48%
No	52.17%	39.53%	64.52%

**Table 5 ijerph-19-15972-t005:** Evaluation of the effect of variables on psychomotor agitation—ordered logistic regression.

**Psychomotor Agitation Total**	**OR**	**95% CI**	** *p* **
Confusional disorder	0.37	0.12 1.11	0.07
Auditory perceptual disorder	4.75	1.19 18.98	0.02 *
**Psychomotor Agitation (Disinhibition Dimension)**	**OR**	**95% CI**	** *p* **
Anxiety	0.53	0.26 1.10	0.09
Confusional disorder	0.27	0.09 0.80	0.01 *
**Psychomotor Agitation (Aggressiveness Dimension)**	**OR**	**95% CI**	** *p* **
Anxiety	1.99	0.98 4.01	0.04 *
Auditory perceptual disorder	3.03	0.88 10.43	0.04 *
**Psychomotor Agitation (Lability Dimension)**	**OR**	**95% CI**	** *p* **
Anxiety	0.39	0.18 0.81	0.01 *

*p*—ordered logistic regression; * statistically significant (*p* < 0.05).

## Data Availability

Please contact the first author.

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
