# Peer review of "Clinical and Sociodemographic Profile of Psychomotor Agitation in Mental Health Hospitalisation: A Multicentre Study"

_ijerph, 2022, doi:10.3390/ijerph192315972_

Round 1

Reviewer 1 Report

The authors assessed clinical and sociodemographic profile of psychomotor agitation in patients with severe mental disorders. The study is relevant as the treatment and handling of patients affected by psychomotor agitation presents major difficulties for health care professionals. Better understanding of associated factors is beneficial for patients and the professionals.

Comments:

1. Section introduction is well written, but it would benefit if the role of nurses and of nursing diagnosis in dealing with patients with psychomotor agitation is mentioned. Specially, since the predictive variables used in analyses refer to NANDA. At the moment, the decision to use nursing diagnosis comes as an arbitrary choice. In section materials and methods it is stated that chosen nursing diagnosis are the ones most closely related to psychomotor agitation, yet this is not mentioned in the Introduction.

2. Section Data, Instrument and Sample would also benefit if few more words were spared explaining the chosen nursing diagnosis and/or the assessment of those.

3. Statistical analysis. I would not have recommend using ordered logistic regression in this case, as it is rather difficult to interpret. However, if choosing to do so please give specifics on planned analysis, such as the name(s) of outcome variables and how many levels they have.

3. Section Results. Table 2. is redundant, it can be described in text with the range of average means and means and SD’s of already mentioned items.

4. Results. The major issue is the interpretation of ordered logistic regression and it should be improved. Consider using term effect instead of influences. Ordered logistic regression does not test correlations per se.

5. Results: Please use uniformed names for ABS total score and its subscales. In Table 5. It is Factor 1 Disinhibition while in Table 6 it is Psychomotor agitation (disinhibition factor). Title of Table 6. should be changed.

6. Discussion:

Line 272-. Who’s study linked the highest level of agitation, yours or that of Hart et al.? It is unclear.

Line 275. add of admitted patients who had an episode of psychomotor agitation. This way it seems as if all admitted patients had psychomotor agitation

1. Line 36, consider replacing "patients suffering from it" with "those affected"

2. Line 90, ...which provides...

3. Lines 129 - 131, ...replace "scores of between" with "scores between"

4. Line 138, gap between aggressiveness and comma

5. Line 142, reformulate the sentence after "....by the ABS scale,". There seems to be a word left over.

6. Line 151, consider using following sentence instead "In addition, median, asymmetry, and kurtosis, were used to describe responses to items the overall score on ABS scale.

7. Table 1., first row there is "n" missing and % is in the wrong column.

8. Line 214, the sentence should be rewritten and simplified, it implies that the number of episodes per patient will be presented. Consider using something like … “As presented in Table 4., majority of patients (64.29%) experienced moderate psychomotor agitation, and 26.42 experienced….”.

Reviewer 2 Report

Dear authors

I find your manuscript interesting in that it deals with an understudied topic, it is clear that psychomotor agitation is a mental health concern, however, I find several areas of opportunity in your manuscript, mainly related to your presentation that can be improved so that the reader understands your objectives and is interested in the article. The authors should be more explicit with the information presented. My comments are as follows

-       I believe that the introduction is well written and presents psychomotor agitation as an important problem for mental health professionals, however, I do not find within the introduction the elements that support the objective of the study. The authors themselves highlight previous findings on the demographic, psychological, and clinical factors associated with psychomotor agitation, so they appear to be already well known: “Psychomotor agitation is associated with certain risk factors [1]. Demographic factors include being male, being aged younger than 40 years, being single, having a family history of alcoholism or aggressive behaviour, having a low level of education and being of a low socioeconomic level; psychological factors include having a history of conflict with healthcare personnel or other patients, recent stressful life events or involuntary or prolonged admission to hospital; and clinical factors include having a family history of previous episodes of agitation, anxiety, fear, substance abuse, low cooperation in treat ment, low-level awareness of illness, cognitive and behavioural disorganisation and positive symptoms, mental retardation, dementia, epilepsy, schizophrenia, and comorbidity with personality disorders” (rows 51-60). Therefore, it is not clear in the introduction what is the innovative element in your research, what is your contribution to the field on this topic.

-       Please include the value of Cronbach's alpha coefficient obtained in your study sample instead of only mentioning it indirectly (rows 133-140).

-       In the material and method section, it is not clear what sociodemographic data were collected. Nor is it specified which clinical factors were considered. I recommend that the authors provide greater clarity and include more information on these aspects.

-       It seems to me that the statistical analyses reveal more ambitious objectives than just describing the sociodemographic and clinical characteristics of a group of patients, so I recommend reformulating the objective so that it better reflects the work performed. As it is written now it may seem of little interest within the field.

-       In the conclusion authors refer: “The clinical and sociodemographic profile of psychomotor agitation in our sample suggests symptoms of moderate/sever level psychomotor agitation in mental health patients.” (rows 311-313). However, this statement does not contribute anything new to the scientific literature on the subject since it is based on the knowledge that mental health problems are associated with psychomotor agitation.

-       In the limitations and strengths the authors state that a strength was “One of the main strengths of this study is its implication for health professionals in the standardisation of an assessment method for episodes of psychomotor agitation in mental health hospital units.” (rows 323-325). However, up to this point the development of an assessment method for psychomotor agitation episodes had not been presented as an objective of the study, but only to describe patients with mental illness and psychomotor agitation

-       The authors could elaborate on the implications of their results on clinical practice as well as on the benefits for society, families with mentally ill relatives or whether these findings could have any political repercussions, considering that as mentioned, there is some stigma around patients with mental illnesses

Round 2

Reviewer 1 Report

Dear authors,

thank you for incorporating previous suggestions. I have no further comments on the paper itself and believe it is a solid one. I appreciate the emphasis on nursing care.

However, there are some language related issues which need to be corrected/improved in the text you added after first reviews. Parts of the text that are problematic are marked with yellow and possible correction are highlighted with green or visible through track changes. Please note that the green suggestions are in line with my understanding of what you wanted to say. You do not have to agree with the proposal, but the ambiguities marked with yellow need to be addressed.

please find the document with suggestions attached

Author Response

Dear reviewer;

Thank you very much for your contributions, I have made the changes according to your indications, as well as, I have uploaded the file again, you will see the changes indicated in green.

Best regards.
